# Latent Variable Causal Discovery under Selection Bias

**Haoyue Dai** [1 2]   **Yiwen Qiu** [1]   **Ignavier Ng** [1]   **Xinshuai Dong** [1]   **Peter Spirtes** [1]   **Kun Zhang** [1 2]

## Abstract

Addressing selection bias in latent variable causal discovery is important yet underexplored, largely due to a lack of suitable statistical tools: While various tools beyond basic conditional independencies have been developed to handle latent variables, none have been adapted for selection bias. We make an attempt by studying rank constraints, which, as a generalization to conditional independence constraints, exploits the ranks of covariance submatrices in linear Gaussian models. We show that although selection can significantly complicate the joint distribution, interestingly, the ranks in the biased covariance matrices still preserve meaningful information about both causal structures and selection mechanisms. We provide a graph-theoretic characterization of such rank constraints. Using this tool, we demonstrate that the one-factor model, a classical latent variable model, can be identified under selection bias. Simulations and real-world experiments confirm the effectiveness of using our rank constraints.

## 1 Introduction

At the core of understanding complex systems lies causal discovery, the identification of causal relations from observational data (Spirtes et al., 2000; Pearl, 2009). In many real-world scenarios, the variables of interest are latent constructs that cannot be directly observed or quantized, while the observed variables serve merely as indirect measurements. For instance, in psychological or political-economic surveys, measured responses serve as proxies for latent personality traits or political orientations. Recovering the causal structure among these latent variables—referred to as latent variable causal discovery—is essential for understanding and reasoning, yet remains a challenging task.

Furthermore, a typical assumption in causal discovery,

---

[1]Carnegie Mellon University [2]Mohamed bin Zayed University of Artificial Intelligence.

*Proceedings of the 42$^{nd}$ International Conference on Machine Learning*, Vancouver, Canada. PMLR 267, 2025. Copyright 2025 by the author(s).

whether involving latent variables or not, is the data being randomly sampled from the underlying population. In practice, however, this is often violated due to selection bias—preferential inclusion of data points based on unknown mechanisms (Heckman, 1977). Returning to the earlier examples, individuals with certain traits may be more willing to take a psychological survey, and methods like mail or phone used to recruit respondents can systematically skew groups based on factors like economic and education level. Ignoring such bias can severely distort the inferred causal structures. Moreover, uncovering the selection mechanisms is also crucial for understanding the data. Hence, there is a pressing need for methods of latent variable causal discovery that can address selection bias.

Despite its importance, addressing selection bias in latent variable causal discovery remains almost unexplored, to the best of our knowledge. One may first recall the Fast Causal Inference (FCI) algorithm (Spirtes et al., 1999), which indeed exploits the conditional independence (CI) constraints in data under both hidden confounding and selection bias. However, FCI is typically not regarded as a method of latent variable causal discovery, as it focuses solely on causal relations among observed variables, with no intension or capability to identify those among latent variables. In short, though FCI can handle both hidden confounding and selection bias, and is already maximally informative under nonparametric CI constraints (Richardson & Spirtes, 2002; Zhang, 2008), it is still not informative enough for latent variable causal discovery. Therefore, new statistical tools that go beyond CI constraints must be developed.

Many new tools beyond CI constraints have thus been developed, typically by imposing additional parametric assumptions. These include rank constraints (Sullivant et al., 2010), equality constraints (Drton, 2018), high-order moment constraints (Xie et al., 2020; Adams et al., 2021; Dai et al., 2022; Chen et al., 2024), constraints based on matrix decomposition (Anandkumar et al., 2013), copula models (Cui et al., 2018), and mixture oracles (Kivva et al., 2021). A detailed review of these tools and the latent variable causal discovery algorithms based on them is provided in Appendix B.

However, all these new tools were developed for latent variables solely, with none adapted to selection bias. While various parametric models for selection were also studied, their

focus is either causal inference (Bareinboim & Pearl, 2012; Correa et al., 2019) or bivariate orientation (Zhang et al., 2016; Kaltenpoth & Vreeken, 2023). For causal discovery, the only tool currently available is still basic CI constraints.

This creates a huge gap: while more powerful latent causal discovery methods now exist, once selection bias is introduced, these newer tools must be set aside, leaving us with only CI constraints—and effectively reverting back to FCI.

Bridging this gap is the goal of our work. We aim to develop tools that go beyond CI constraints and address both latent variables and selection bias. The core challenge lies in modeling data under selection, which can be far more complex than handling latent variables. For instance, marginalizing a linear Gaussian model over latent variables still yields a Gaussian distribution with a closed-form covariance in terms of model parameters. However, under selection, even simple truncation leads to a truncated Gaussian distribution, making its covariance and higher moments hard to express and interpret (Kan & Robotti, 2017). This situation only worsens with more complex selection mechanisms.

To address this challenge, we try to avoid explicitly modeling the full distribution under selection and instead focus on invariant statistical patterns, much like nonparametric CI constraints with the corresponding d-separation graphical criterion. This leads us to rank constraints, a direct generalization to the CI constraints, which assumes data generated by a linear Gaussian model and exploits the (low) ranks of covariance submatrices. In this case, CIs correspond to zero partial correlations, which manifest as low ranks in covariance submatrices. There are also other low ranks beyond zero partial correlations. Rank constraints captures them with graphical criterion beyond d-separation, that is, *t-separation*. Details and examples are provided in §2.1.

But, do rank constraints remain informative in selection-biased data, where the data no longer follows a linear Gaussian model—or even a linear structural equation model? Interestingly, the answer is yes. For data generated by a linear Gaussian model and subjected to selection, even though the biased covariance matrix becomes arbitrarily complex to express, we show that the ranks of these covariances remain well-defined and capture meaningful structural information about both causal and selection mechanisms. Illustrative examples are provided in §2.2. Specifically, we assume *linear selection mechanisms*, which will be formally defined later. For now, let us note that it is general, with many existing parametric selection models as special instances.

As shown in Figure 1, the main contribution of this work is the *generalized rank constraints*, which, to the best of our knowledge, is the first tool beyond CI constraints to handle selection, and thus enables latent variable causal discovery under selection bias. Just as the original rank constraints

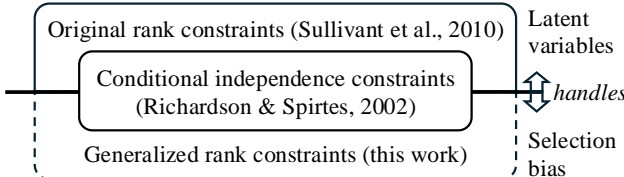

*Figure 1.* Illustration of our contribution: while CI constraints handles both latent variables and selection bias, original rank constraints extends it only to latent variables. We bridge this gap by generalizing rank constraints to handle also selection bias.

generalized CI constraints and enabled algorithms beyond FCI, our generalized rank constraints pave the way for algorithms to uncover both latent causal and selection structures.

The remainder of this paper is organized as follows. In §2, we introduce original rank constraints in settings without selection bias, and then provide an illustrative example showing how ranks can retain information under selection. In §3, we formally characterize the generalized rank constraints under selection bias, presenting a precise graphical criterion. We show that these ranks offer insights into both latent causal and selection structures. In §4, we apply the generalized rank constraints to the one-factor model, a classical latent variable model, and demonstrate its identifiability under selection bias. In §5, we validate the effectiveness of our method through simulations and real-world experiments, showing its ability to uncover latent causal and selection mechanisms. Finally, in §6, we discuss potential limitations.

**Notations on matrices.** For a matrix $M$, we let $M_{i,j}$ be its $(i,j)$-th entry. For two index sets $A, B$, we let $M_{A,B} = (M_{a,b})_{a \in A, b \in B}$ be the submatrix of $M$ with rows indexed by $A$ and columns indexed by $B$. For a finite set $A$, we denote by $|A|$ the cardinality of $A$.

**Notations on graphs.** In a directed acyclic graph (DAG) $\mathcal{G}$, for any vertices $a, b$, we say $a$ is a *parent* of $b$ and $b$ is a *child* of $a$ if $a \to b$ is an edge in $\mathcal{G}$, denoted by $a \in \mathrm{pa}_{\mathcal{G}}(b)$ and $b \in \mathrm{ch}_{\mathcal{G}}(a)$; $a$ is an *ancestor* of $b$ and $b$ is a *descendant* of $a$ if $a = b$ or there is a directed path $a \to \cdots \to b$ in $\mathcal{G}$, denoted by $a \in \mathrm{an}_{\mathcal{G}}(b)$ and $b \in \mathrm{de}_{\mathcal{G}}(a)$. These notations extend to sets: e.g., for any vertex set $A$, $\mathrm{an}_{\mathcal{G}}(A) := \bigcup_{a \in A} \mathrm{an}_{\mathcal{G}}(a)$.

## 2. Motivation

In this section, we provide the background and motivation behind our approach. In §2.1, we review the basics of the original rank constraints without selection bias and illustrate how latent variables can leave traces in the covariances among observed variables. Building on this intuition, in §2.2, we provide an illustrative example showing how selection may also leave traces in the biased covariances.

### 2.1. Preliminaries on Original Rank Constraints

Let us provide the background to and definitions of the rank

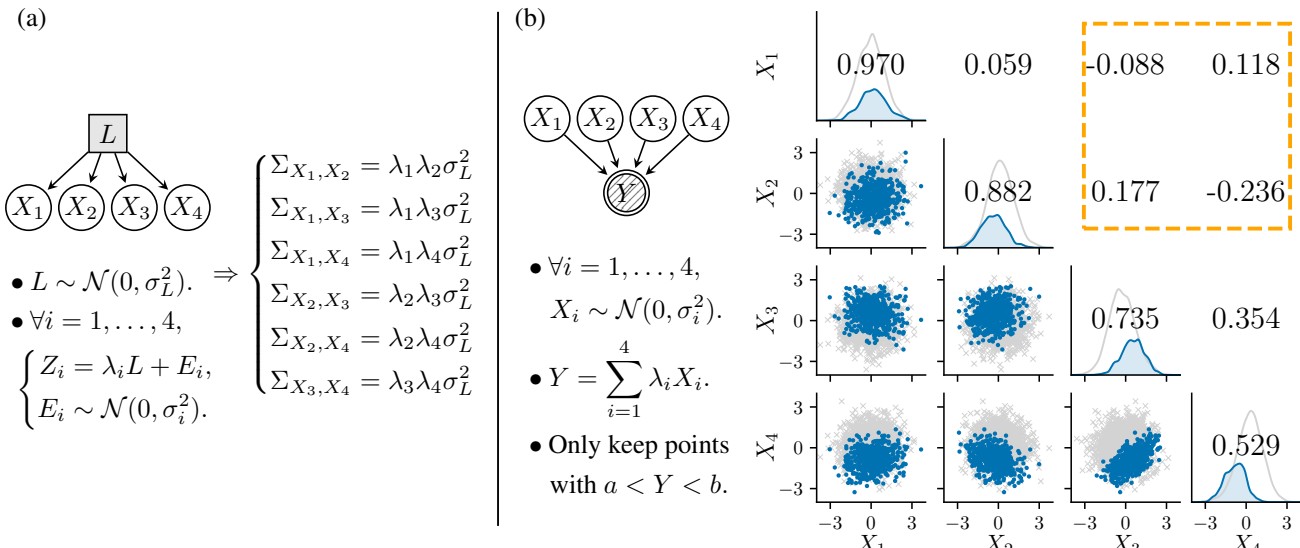

*Figure 2.* Illustrative examples of rank constraints. White circles are observed variables. Grey squares are latent variables. Lined double circles are selection response variables (i.e., only samples with specific values of them are collected in data). We use this color scheme throughout. (a) shows the original Tetrad structure from Spearman (1914). The covariance terms represented by model parameters are given. Since the distribution is simply joint Gaussian, their scatterplot is omitted. (b) shows an "inverse Tetrad structure," where four originally independent variables are later truncated based on a linear sum of them. An example with $\sigma_i^2 = 1$, $\lambda_1, \lambda_2, \lambda_3, \lambda_4 = 1, -2, 3, -4$, and $a, b = 3, 10$ is visualized in the scatterplot, showing both selected samples ('●') and unselected samples ('×'), alongside the covariance values in the selected data ('●'). The dashed box highlights one of the low-rank structures.

constraints and its corresponding t-separation graphical criterion, as originally established in (Sullivant et al., 2010). At this stage, selection bias is yet to be introduced.

We consider a linear Gaussian causal model associated with a DAG $\mathcal{G}$, in which random variables $X = (X_1, \ldots, X_m)$ follow the data generating procedure:

$$X = \Lambda X + E, \tag{1}$$

where $E$ are exogenous noise terms that follow a jointly independent Gaussian distribution, and $\Lambda$ is the weighted adjacency matrix that follows $\mathcal{G}$, i.e., $X_i \to X_j$ is an edge in $\mathcal{G}$ if $\Lambda_{j,i} \neq 0$. Let $\Phi$ be the diagonal covariance matrix of noise terms $E$. The covariance matrix of Gaussian variables $X$, denoted by $\Sigma$, can then be written as

$$\Sigma = (I - \Lambda)^{-1} \Phi (I - \Lambda)^{-\top}. \tag{2}$$

Rank constraints investigates the ranks of submatrices in this covariance matrix $\Sigma$. It is first shown to have generalized conditional independencies as a special case:

**Proposition 1** (Conditional independencies as low ranks; Prop. 2.2 in Sullivant et al. (2010)). *Let $A, B, C$ be disjoint subsets of $X$. Then the conditional independence $A \perp\!\!\!\perp B | C$ holds, if and only if the submatrix $\Sigma_{A \cup C, B \cup C}$ has rank $|C|$.*

Proposition 1 is a direct reformulation of the corresponding zero partial correlations, which, since $X$ are Gaussian, manifest as conditional independencies. However, there are also other low ranks beyond CI, as the following example shows.

**Example 1.1** (Tetrad structure in Spearman (1914)). Consider the graph in Figure 2a. In this graph, for any choice of model parameters, the following three low ranks hold:

$$\text{rank}(\Sigma_{\{X_1, X_2\}, \{X_3, X_4\}}) = 1,$$
$$\text{rank}(\Sigma_{\{X_1, X_3\}, \{X_2, X_4\}}) = 1,$$
$$\text{rank}(\Sigma_{\{X_1, X_4\}, \{X_2, X_3\}}) = 1,$$

as can be verified from the covariance terms provided in the figure. However, these low ranks do not follow from any CIs. In fact, there are no CIs among $\{X_1, X_2, X_3, X_4\}$. △

Rank constraints explains where these extra low ranks come from. Similar to how Pearl & Verma (1987) used the d-separation graphical criterion to characterize CIs entailed in data, Sullivant et al. (2010) characterized these entailed low ranks using a new criterion, namely, t-separation:

**Definition 1** (t-separation; reformulated from Def. 2.7 in Sullivant et al. (2010)). Let $A, B, C, D$ be four subsets of $X$ that need not be disjoint. Denote by $\mathcal{G}_{\overline{C}}$ the remaining graph after removing nodes $C$ and their associated edges from $\mathcal{G}$, and similarly $\mathcal{G}_{\overline{D}}$. We then say the pair $(C, D)$ t-separates $(A, B)$ if $\text{an}_{\mathcal{G}_{\overline{C}}}(A \backslash C) \cap \text{an}_{\mathcal{G}_{\overline{D}}}(B \backslash D) = \varnothing$.

Roughly speaking, to find node sets that t-separate $A$ from $B$, we are to find node sets whose removal from the graph makes $A$ and $B$ share no common ancestors, so that all pathways that carry shared information are blocked. Ranks in covariances reflect exactly the smallest size of such sets:

**Proposition 2** (Graphical criterion of rank constraints; Thm. 2.8 in Sullivant et al. (2010)). *In a DAG $\mathcal{G}$, for any two subsets $A, B \subset X$ that need not be disjoint, the equality*

$$\text{rank}(\Sigma_{A,B}) = \min\{|C|+|D| : (C, D) \text{ t-separates } (A, B)\}$$

*holds for generic choice of model parameters.*

Regarding the term *generic*, equality in Proposition 2 holds for almost all parameter choices, except for a set with Lebesgue measure 0 where coincidental lower ranks occur. Hereafter, when we say "assuming genericity", we exclude such coincidental cases, similar to how standard *faithfulness* assumes no CIs other than those entailed by d-separations.

Let us examine this criterion in the Tetrad example:

**Example 1.2** (t-separation in Tetrad structure). Continue on the graph shown in Figure 2a. We have that $(\varnothing, \{L\})$ or $(\{L\}, \varnothing)$ t-separate $(\{X_1, X_2\}, \{X_3, X_4\})$, which explains the first $\text{rank} = 1 \ (= 0 + 1)$ in Example 1.1. The other two low ranks can be explained in a similar way. $\triangle$

Rank constraints and its t-separation criterion directly generalize the CI constraints and its d-separation criterion, and thus offer more structural insights into latent variables than those given by FCI. We illustrate with the Tetrad example:

**Example 1.3** (Rank constraints enables latent variable identification). Continue on the graph in Figure 2a. Suppose that for some reason $L$ is latent, leaving only $X_1, X_2, X_3, X_4$ observed. Using CI constraints alone, algorithms like FCI cannot distinguish this model from an alternative fully connected graph with the same four observed variables, as no CIs exist in both models. However, with rank constraints, this alternative is falsified, as otherwise the three low ranks cannot be satisfied. Ranks reveal that though there is no conditional independence, the dependence among data is not arbitrary—it must stem in a single-dimensional way, suggesting the presence of latent variables. $\triangle$

Based on this insight, various latent variable causal discovery methods have been developed (reviewed in Appendix B). However, as noted, none of them address selection bias.

## 2.2 Ranks in Selection-Biased Data: an Example

We now present an example to illustrate how the ranks of covariances may remain informative about the causal structure, even for the data under selection bias.

Let us first revisit the motivation behind the original rank constraints without selection bias, as introduced in §2.1. The proof of the t-separation criterion consists of two key components: it first uses algebraic combinatorial tools to interpret covariance terms (which are polynomials of model parameters, as in Figure 2a) as the "flows of information" in the graph, and then uses the max-flow–min-cut principle

to count for the smallest size of nodes needed to "choke" all these flows. Simply put, when some dependence in the data cannot be fully explained (i.e., rendered conditionally independent), rank constraints can serve to quantify the "dimensional bottleneck" of how this dependence stems from.

The question then arises: can this "dimensional bottleneck" still be reflected in ranks, when the selection mechanism is "low-dimensional"? We consider the following example.

**Example 2.1** (Inverse Tetrad structure). Consider the graph shown in Figure 2b, which we call the "inverse Tetrad structure". We model a simple truncation selection mechanism, where the four observed variables $X_1, X_2, X_3, X_4$ are originally mutually independent, but then get selected by truncating on the value of a linear sum of them, represented by the unobserved "response" variable $Y$ in the graph.

This simple truncation can introduce many complexities: as seen in the scatterplot, the remaining population no longer follows a linear Gaussian model, or even a linear structural equation model, that is, no independent exogenous noise terms can be found to generate the data. Instead, the data follows a truncated Gaussian distribution, where, unlike the closed-form polynomials in Figure 2a, the covariances and higher moments are difficult to express. Hence, we show the covariance values in a specific simulation in the scatterplot.

Yet, an interesting observation emerges from these values. Despite the complexities introduced by selection, the low-rank structure of covariances seems preserved. Specifically, they are the same as in the original Tetrad structure. Denote by $\Sigma'$ the covariance matrix of the biased data, we have:

$$\text{rank}(\Sigma'_{\{X_1, X_2\}, \{X_3, X_4\}}) = 1,$$
$$\text{rank}(\Sigma'_{\{X_1, X_3\}, \{X_2, X_4\}}) = 1,$$
$$\text{rank}(\Sigma'_{\{X_1, X_4\}, \{X_2, X_3\}}) = 1,$$

as can be verified from the values in the scatterplot (the first one is already highlighted in the dashed box). $\triangle$

Example 2.1 strongly suggests that selections, just like latent variables, may leave identifiable traces about their "dimensional bottleneck" in the ranks of even biased data. Questions naturally arise: What if the selection becomes more complex (e.g., involving randomness, unlike truncation)? What if multiple selection mechanisms act simultaneously? When low ranks occur, can we differentiate whether it is from latent variables or selection bias? We formalize and aim to answer these questions in the next section.

## 3 Generalized Rank Constraints

In this section, we formalize the generalized rank constraints for handling selection bias. In §3.1, we define the linear selection mechanism, a model both general and necessary for rank constraints to work. In §3.2, we provide the graphical

criterion for ranks in the biased covariances. Finally, in §3.3, we discuss the identifiability of distinguishing between latent variables and selection bias using rank constraints.

## 3.1 Linear Selection Mechanisms

In this part, we introduce the linear selection mechanism. Similar to how original rank constraints generalize nonparametric CI constraints within linear Gaussian models, in this paper, we focus on a specific class of selections called linear selection mechanisms, defined below.

**Definition 2** (Linear selection mechanism). For a set of variables $X$, a linear selection mechanism is described by a configuration $\mathcal{S}$, which consists of tuples $\{(V_i, \beta_i, \epsilon_i, \mathcal{Y}_i)\}_{i=1}^k$. Each tuple specifies a single selection condition, where:

- $V_i \subseteq X$ is the subset of variables from $X$ directly involved in the $i$-th selection;
- $\beta_i \in \mathbb{R}_{\neq 0}^{|V_i|}$ is a vector of nonzero linear coefficients that specifies how variables in $V_i$ contribute to the selection;
- $\epsilon_i$ is an independent noise term that models selection randomness. It may follow an arbitrary distribution, including non-Gaussian, or be degenerate to a constant;
- $\mathcal{Y}_i \subsetneq \mathbb{R}$ is the set of admissible values, a proper subset of $\mathbb{R}$, which may consist of a single value, multiple values, an interval, or a union of intervals, etc.

For each single selection, we call $Y_i = \beta_i^\top V_i + \epsilon_i$ as its response variable. Finally, a sample of $X$ is included in the selected data if and only if $Y_i \in \mathcal{Y}_i$ for all $i = 1, \ldots, k$.

The linear selection mechanism is versatile, allowing multiple selections to act simultaneously, as commonly seen in real world like multi-criteria admissions. Each single selection is also flexible, with various existing parametric models for selection bias fitting as specific instances:

**Example 3** (Existing parametric models fit in the linear selection mechanism). Let us first explain what is modeled by a single linear selection. For a sample $X$, the probability for it to be selected in the $i$-th selection is:

$$P(Y_i \in \mathcal{Y}_i \mid X) = \int_{\mathcal{Y}_i - \beta_i^\top V_i} p_{\epsilon_i}(u) \, du. \tag{3}$$

where $\mathcal{Y}_i - \beta_i^\top V_i = \{u \in \mathbb{R} : u + \beta_i^\top V_i \in \mathcal{Y}_i\}$. Then, with different choices of $\epsilon_i$ and $\mathcal{Y}_i$, many existing common selection models can fit as instances, including:

- $\epsilon_i = 0$ and $\mathcal{Y}_i = (a, b)$ reduce to a hard truncation model, as illustrated in Figure 2b;
- $\epsilon_i \sim \text{Logistic}(0, 1)$ and $\mathcal{Y}_i = (a, \infty)$ reduce to a logistic selection model (Dubin & Rivers, 1989);
- $\epsilon_i \sim \mathcal{N}(0, 1)$ and $\mathcal{Y}_i = (a, \infty)$ reduce to a probit selection model (Heckman, 1977);
- $\epsilon_i \sim \mathcal{N}(0, 1)$ and $\mathcal{Y}_i = \{a\}$ reduce to a stabilizing selection model (Lande & Arnold, 1983). $\triangle$

Having defined the linear selection mechanism and demonstrated its connection to existing models, we now turn to the graphical criterion of rank constraints under selection.

## 3.2 Graphical Criterion of Ranks under Selection

We are now ready to provide the graphical characterization of the covariance ranks in selection-biased data.

Since selection can also be viewed as a causal process in the data generating procedure, we incorporate it into the causal graph. This leads to the definition of the selection-augmented graph, following Bareinboim & Pearl (2012):

**Definition 3** (Selection-augmented graph). Consider a DAG $\mathcal{G}$ with nodes $X$ that represents the original data generating process for $X$, and a linear selection configuration $\mathcal{S} = \{(V_i, \beta_i, \epsilon_i, \mathcal{Y}_i)\}_{i=1}^k$ with $k$ single selection mechanisms. The selection-augmented graph is a new DAG, denoted $\mathcal{G}^{(\mathcal{S})}$, obtained by augmenting $\mathcal{G}$ with the following:

- Additional selection response nodes $Y = \{Y_i\}_{i=1}^k$, and
- Additional edges $\{X_j \to Y_i : \forall i = 1, \cdots, k, X_j \in V_i\}$.

The post-selection data $X$ can then be viewed as generated from $\mathcal{G}^{(\mathcal{S})}$, with each response $Y_i$ restricted within its admissible values $\mathcal{Y}_i$. When there is no selection, i.e., $\mathcal{S} = \varnothing$, the $\mathcal{G}^{(\varnothing)}$ reduces to the original $\mathcal{G}$.

Note that, unlike conventional notation, we use the letter "$Y$" instead of "$S$" to denote selection variables. This is because, conventionally, selection refers to conditioning on a single value (often Boolean in nonparametric settings) of a variable, whereas here we allow $Y_i$ to take multiple values.

With the selection-augmented graph, we now provide the graphical criterion for generalized rank constraints:

**Theorem 1** (Graphical criterion for generalized rank constraints). *Let $\mathcal{G}$ be a DAG and $X$ the variables generated from $\mathcal{G}$ with a linear Gaussian model as specified in Equation (1). Suppose $X$ then undergoes linear selections specified by $\mathcal{S} = \{(V_i, \beta_i, \epsilon_i, \mathcal{Y}_i)\}_{i=1}^k$ with $k$ single selections. Let $\Sigma^{(\mathcal{S})}$ be the population covariance matrix of $X$ after selection. For any two subsets $A, B \subset X$ which need not be disjoint, assuming genericity, we have:*

$$\text{rank}(\Sigma_{A,B}^{(\mathcal{S})}) = \min\{|C| + |D| : C, D \subset X \cup Y,$$
$$(C, D) \text{ t-separates } (A \cup Y, B \cup Y) \text{ in } \mathcal{G}^{(\mathcal{S})}\} - k,$$

*where $Y$ denotes the additional selection response variables introduced to the selection-augmented graph $\mathcal{G}^{(\mathcal{S})}$.*

The proof of Theorem 1 is provided in Appendix A. Note that when there is no selection (i.e., $\mathcal{S} = \varnothing$ and $k = 0$), the theorem reduces to Proposition 2. That is, our graphical criterion generalizes the original rank constraints to accommodate selection bias, hence the name "generalized rank

*Table 1.* The spider example and its variants with selection bias. In the selection-augmented graphs, each node represents a group of non-adjacent variables (e.g., $C$), with the lowercase letter indicating its cardinality (e.g., $c = |C|$). Edges between nodes represent fully connected directed edges from every variable in one group to every variable in another. We assume $a, b \gg l, r > c, d$. The table presents the rank values (upper rows) and corresponding t-separation sets (lower rows) among observed variables in $A, B$, derived from Theorem 1. Here, $A_1, A_2$ denote large enough disjoint subsets of $A$, and $B_1, B_2$ similarly. For brevity, unions like $A_1 \cup B_1$ are written as $A_1 B_1$. For example, in the second graph, the rank of the covariance matrix between $A$ and $B$ is $2c + d$, with $(CD, CD)$ t-separating $(AD, BD)$.

| Graphs and covariance submatrices | | | | |
|---|---|---|---|---|
| $A_1, A_2$ | $l + c$ $(\varnothing, LC)$ | $l + c$ $(\varnothing, LCD)$ | $l + c$ $(LR, LC)$ | $l + c$ $(LRC, LC)$ |
| $B_1, B_2$ | $r + c$ $(\varnothing, RC)$ | $r + c$ $(\varnothing, RCD)$ | $r + c$ $(LR, RC)$ | $r + c$ $(LRC, RC)$ |
| $A, B$ | $2c$ $(C, C)$ | $2c + d$ $(CD, CD)$ | $2c$ $(RC, LC)$ | $c$ $(RC, LC)$ |
| $A_1 B_1, A_2 B_2$ | $l + r + c$ $(LC, R)$ | $l + r + c$ $(LC, RD)$ | $l + r$ $(LR, LR)$ | $l + r + c$ $(LRC, LRC)$ |

constraints". This bridges the gap beyond basic CI constraints to handle selection bias, as illustrated in Figure 1.

Let us examine this criterion in the inverse Tetrad example:

**Example 2.2** (t-separation in inverse Tetrad structure)**.** Let us revisit the model in Example 2.1. The corresponding selection-augmented graph is shown in Figure 2b. In it, we have that $(\{Y\}, \{Y\})$ t-separates $(\{X_1, X_2, Y\}, \{X_3, X_4, Y\})$, which explains the first $\text{rank} = 1 (= 1 + 1 - 1)$ in Example 2.1. The other two low ranks can be explained in a similar way. △

Generalized rank constraints reveal that when dependence in the data cannot be fully conditioned out, the "dimensional bottleneck" of how this dependence stems from–be it latent variables or selection bias–can leave traces in ranks of covariance matrix of even biased data. This provides a powerful graphical tool for identifying causal structures that involve both latent variables and selection bias.

Before applying this tool for latent variable causal discovery, a question arises. Recall the Tetrad and inverse Tetrad structures in Examples 1.1 and 2.1. Though one has latent variables and the other has selection bias, their rank constraints among four observed variables are identical. That is, while ranks reveal a single-dimensional bottleneck in the observed dependence, they cannot distinguish whether this arises from latent variables or selection bias. Then, does every selection structure have an alternative structure—free of selection but may involving latent variables—that is equivalent in ranks? We explore this in the next part.

### 3.3 Identifiability between Latent Variables and Selection Bias: Examples

Now, we discuss the possibility of distinguishing between latent variables and selection bias, using rank constraints from data that may or may not be selected.

For two selection-augmented DAGs over the same observed variables $X$, but different and possibly empty latent variables and selection response variables, we say they are "rank equivalent" if their covariance submatrices for any $A, B \subset X$ have identical ranks. This parallels the concept of "CI equivalence", where, consider for example two measured variables, whether they are confounded $(X_1 \leftarrow L \rightarrow X_2)$, directly causal related $(X_1 \rightarrow X_2)$, or selected $(X_1 \rightarrow Y \leftarrow X_2)$ are indistinguishable by CI, as all three have $X_1 \not\perp X_2$. We then answer the following question: for every graph with selection, is there always an alternative graph–free of selection but may involving latent variables—that is rank equivalent to it, and vice versa?

Interestingly, the answer is no, contrary to the intuition suggested by Examples 1.1 and 2.1. We first note that latent variables and selection bias can sometimes be distinguished using only CI constraints, as shown below.

**Example 4** (CI constraints to distinguish between latent and selection variables)**.** Consider the following two graphs.

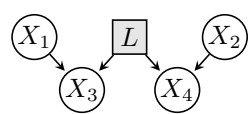 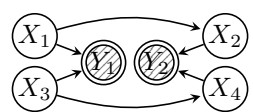

In the first one, with $X_1 \perp\!\!\!\perp X_4$, $X_2 \perp\!\!\!\perp X_3$, $X_1 \not\perp\!\!\!\perp X_4|X_3$, and $X_2 \not\perp\!\!\!\perp X_3|X_4$, it can be concluded that latent variables must exist between $X_3$ and $X_4$. In the second one, with the only two CIs among observed variables being $X_1 \perp\!\!\!\perp X_4|X_2, X_3$ and $X_2 \perp\!\!\!\perp X_3|X_1, X_4$, it follows that selection bias involving all four variables must be present. $\triangle$

These examples are originally introduced as illustrations for the FCI algorithm (Zhang, 2008). The derivation uses v-structure orientation rules, omitted here for brevity.

We then note that beyond CI, rank constraints can sometimes also distinguish between latent variables and selection bias, even in CI equivalent graphs, as shown below.

**Example 5** (Rank constraints to distinguish between latent and selection variables). Consider the graphs in Table 1. The first column shows the original "spider" structure in (Sullivant et al., 2010), and the other three are its variants with selection bias, as detailed in the table caption. In these graphs, only $A$ and $B$ are observed, with no CIs among them. However, low ranks exist and differ across graphs.

In the first column, $A$ and $B$ share latent variables $C$, yet the rank between them is $2|C|$ instead of $|C|$, a property unique to the original spider structure (up to indeterminacies inside groups). In contrast, in the second and third columns, while other ranks remain unchanged, either the rank increases between $A, B$ or decreases between $A_1B_1, A_2B_2$, which can not be achieved by any graph without selection. $\triangle$

Beyond these examples, we have to note that the completeness in distinguishing latent variables from selection bias requires characterizing the rank equivalence class, analogous to maximal ancestral graphs (MAGs) for CI constraints (Richardson & Spirtes, 2002). This remains an open challenge and is beyond the scope of this paper.

Now, having introduced generalized rank constraints and its graphical criterion, let us apply it to latent causal discovery with selection bias in a specific model class.

## 4    One-factor Model under Selection Bias

In this section, we illustrate how the generalized rank constraints helps identify latent causal structure under selection bias. For a case study, we consider the one-factor model.

The one-factor model from Silva et al. (2003) captures cases where latent variables are indirectly measured, such as questionnaires. Since selection bias is also common in such data (e.g., personal traits influencing survey participation), we extend the model to incorporate it, as defined below.

**Definition 4** (One-factor model with selection bias). Let $\mathcal{G}$ be a DAG generating latent variables $L = \{L_1, \ldots, L_m\}$. Each latent variable $L_i$ has measurements $\mathbf{X}_i$ (i.e., children that has no other parents than $L_i$) with $|\mathbf{X}_i| \geq 2$.

The data are further subject to a possible selection $\mathcal{S} = \{(V_j, \beta_j, \epsilon_j, \mathcal{Y}_j)\}_{j=1}^k$ to latent variables, i.e., $\bigcup_{j=1}^k V_j \subset L$.

Using the graphical criterion in Theorem 1, we show that CI among $L$, even unobserved and biased, can still be recovered from the rank constraints of the observed data:

**Proposition 3** (Ranks recover CIs among $L$ under selection bias). *For any disjoint subsets $A, B, C \subset L$, $A \perp\!\!\!\perp B \mid C$ holds in the selection-biased data, if and only if the rank of the population covariance matrix between $\mathbf{X}_A \cup \mathbf{X}_C^{(1)}$ and $\mathbf{X}_B \cup \mathbf{X}_C^{(2)}$ is $|C|$. Here, $\mathbf{X}_C^{(1)}$ and $\mathbf{X}_C^{(2)}$ are disjoint partitions of $X_C$ such that $|\mathbf{X}_C^{(1)}|, |\mathbf{X}_C^{(2)}| \geq |C|$.*

Proposition 3 is a direct consequence from the graphical criterion, showing that the d-separations among $L$ persist as low-rank structures in $X$, even under selection bias. By recovering these CIs, constraint-based algorithms such as FCI can then be applied to $L$ as if $L$ were directly observed. The next section presents experiments that leverage this insight.

## 5    Experiments and Results

Having introduced the generalized rank constraints and proposed method for one-factor model, we conduct experiments on synthetic and real-world data, showing that our method effectively recovers the causal structure under selection.[1]

### 5.1    Experiments on Synthetic Data

We conduct empirical studies on synthetic data to evaluate our method against existing ones. Specifically, we simulate linear SEMs under the truncated probit model. We first generate a random Erdös–Rényi graph (Erdös & Rényi, 1959) among the $n \in \{5, 10, 15, 20\}$ latent variables, with an average degree of 2. Each latent variable has either 2 or 3 observed variables as children. The linear coefficients of the edges are sampled uniformly at random from $[-2, -0.5] \cup [0.5, 2]$. For $n$ latent variables, we simulate $n/5$ selection variables using linear selection mechanisms with error terms, where each selection variable has an average of $\lceil 0.3n \rceil$ parents chosen from the latent variables. We then retain only the samples where these selection variables fall within the 40th to 60th percentile of their values. Here, we consider both Gaussian and exponential distributions for the error terms.

Since the goal is to validate Proposition 3, we evaluate the partial ancestral graph (PAG) (Spirtes et al., 2000) among the latent variables estimated by FCI. Thus, we assume access to the oracle clustering in the one-factor model—that is, knowing which observed variables correspond to the measurements of which latent variables. We compare our method against FCI (Spirtes et al., 2000), PC (Spirtes & Glymour, 1991), and BOSS (Andrews et al., 2023). We include

---

[1]An implementation of our method is available at https://github.com/MarkDana/Latent-Selection.

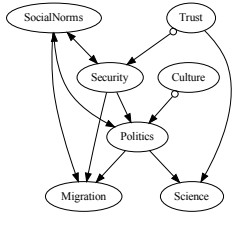

(a) Output PAG in *Canada*

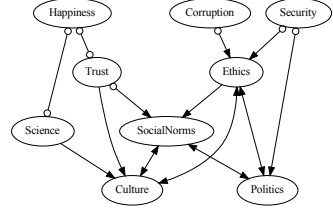

(b) Output PAG in *China*

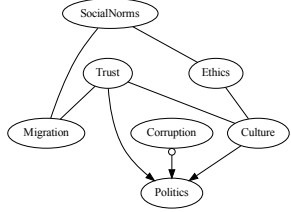

(c) Output PAG in *Germany*

*Figure 3.* Estimated PAGs on the World Value Survey dataset by country. Arrows ($\rightarrow$) indicate ancestral causal relations, double tails ( — ) and double heads ($\leftrightarrow$) suggest selection bias and latent confounding, respectively, and open circles ($\circ$) denote uncertainty in orientation.

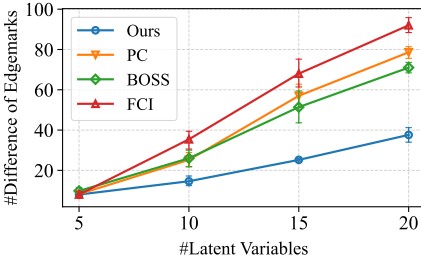

*Figure 4.* Results with Gaussian error terms. The y-axis shows the total number of differing edge marks between the ground-truth PAG and the estimated PAG (lower is better).

BOSS in the comparison as it has been shown to outperform classical score-based methods such as GES (Chickering, 2002). Since these three methods learn structures over observed variables rather than directly modeling the relationships among latent variables, we randomly select one observed child (i.e., measurement) as the representative for each latent variable and use these methods to infer the structure among these chosen representatives. For PC and BOSS, we convert their output to a PAG. To evaluate the estimated structure among the latent variables, we report the differences in edge marks for the estimated PAG and the true one.

The results for Gaussian error terms are presented in Figure 4, while those for exponential error terms are provided in Figures 5 and 7 in Appendix C. Notably, our method performs the best across all settings, with the performance gap widening as the number of variables increases. This demonstrates the effectiveness of our approach in recovering the latent PAG under selection bias and further validates Proposition 3. All experiments are from 5 random runs with 2 CPUs and 16 GB of memory. Each run with a number of latent variables of 5, 10, or 15 requires less than one second, and with 20 latent variables requires less than five minutes.

## 5.2 Experiments on Real-World Data

**Datasets.** In this part, we present experimental results on real-world datasets. We examine the (1) **World Value Survey**[2] (WVS) dataset and the (2) **Big Five Personal-**

ity[3] (BIG5) dataset. Both datasets are in questionnaire format, and thus, can be effectively captured by the one-factor model. As for (1), WVS is the largest non-commercial academic social survey program, organized in waves and conducted every five years. The survey provides time-series data spanning 39 years (1981–2020), includes over 600 indicators (questions), and covers 120 countries, ensuring a global scope. WVS is devoted to the scientific and academic study of social, political, and cultural values of people in the world, nevertheless, the relations between these values remain under-researched. Here, we focus on 259 indicators of 13 core variables of interest. The core variables include `Social Norms`, `Happiness`, `Security Values`, `Ethical Values` and so on. For succinctness, a single keyword is employed for each variable represented in the graph. Data are collected across different nations, with varying sizes ($500 \sim 4000$ samples). Detailed information for the nation-specific data and the preprocessing scheme are provided in Appendix C. As for (2), the five personality dimensions are `Openness`, `Conscientiousness`, `Extraversion`, `Agreeableness`, and `Neuroticism` (O-C-E-A-N), are measured with their own 10 indicators, with a total of 50 questions and approximately 20000 samples. This dataset has been closely examined by Dong et al. (2023), however, the possibility of selection bias has not been considered.

**Results.** Figure 3 presents the results corresponding to the estimated PAG over three countries in different continents: *Canada*, *China*, and *Germany*. Note that some variables are missing in certain countries due to the low response rate. *Finding (A)*: We observe a similar (potential) selection pattern across three countries. In particular, the tail at node `Social Trust` is indicative of the node's potential role as an ancestor of selection. The interpretation is that people's propensity to complete a questionnaire is based on their degree of trust towards others. *Finding (B)*: Different countries also exhibit nation-specific potential selection patterns. As shown in Figure 3b, we can observe a potential selection on `Perception of Science`. This potential selection can be interpreted as follows: a positive perception

[2] https://www.worldvaluessurvey.org/WVSDocumentationWV7.jsp

[3] https://openpsychometrics.org/

of scientific studies is associated with a higher probability of participation in this social science study. Moreover, in Figure 3c, we observe that selection bias involving five variables is present. *Finding (C)*: For the BIG5, a selection based on `Agreeableness` is present, where a hypothesis could be made that higher levels of agreeableness may facilitate higher levels of responsiveness. We leave the result for BIG5 in Appendix C. These findings not only serve as intuitive validation of our main theorems and method, but also help us to recover the whole ground-truth underlying data structure by recognizing the potential selection.

# 6 Conclusion and Limitations

In this work, we develop the generalized rank constraints, providing a tool to identify latent causal structure under selection bias. A limitation is that a complete rank equivalence characterization has yet to be developed.

# Acknowledgment

We would like to acknowledge the support from NSFAward No. 2229881, AI Institute for Societal Decision Making (AI-SDM),the National Institutes of Health (NIH) under Contract R01HL159805, and grants from Quris AI, Florin Court Capital, and MBZUAI-WIS Joint Program. IN acknowledges the support of the Natural Sciences and Engineering Research Council of Canada (NSERC) Postgraduate Scholarships – Doctoral program.

# Impact Statement

This paper presents work whose goal is to advance the field of causal discovery. Our method has implications for analyzing e.g., personality traits, political attitudes, and cultural differences in social science related datasets. While our work enhances such tasks, responsible application is crucial to avoid misinterpretation on selections in the results.

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

# A   Proofs of Main Results

**Theorem 1** (Graphical criterion for generalized rank constraints). *Let $\mathcal{G}$ be a DAG and $X$ the variables generated from $\mathcal{G}$ with a linear Gaussian model as specified in Equation* (1). *Suppose $X$ then undergoes linear selections specified by $\mathcal{S} = \{(V_i, \beta_i, \epsilon_i, \mathcal{Y}_i)\}_{i=1}^k$ with $k$ single selections. Let $\Sigma^{(\mathcal{S})}$ be the population covariance matrix of $X$ after selection. For any two subsets $A, B \subset X$ which need not be disjoint, assuming genericity, we have:*

$$\mathrm{rank}(\Sigma_{A,B}^{(\mathcal{S})}) = \min\{|C| + |D| : C, D \subset X \cup Y,$$
$$(C, D) \text{ t-separates } (A \cup Y, B \cup Y) \text{ in } \mathcal{G}^{(\mathcal{S})}\} - k,$$

*where $Y$ denotes the additional selection response variables introduced to the selection-augmented graph $\mathcal{G}^{(\mathcal{S})}$.*

*Proof of Theorem* 1. To prove this graphical criterion, we are to show an equality between ranks in selected and unselected covariances:

$$\mathrm{rank}(\Sigma_{A,B}^{(\mathcal{S})}) = \mathrm{rank}(\Sigma_{A \cup Y, B \cup Y}) - |Y|,$$

where $\Sigma$ denotes the original global joint covariance matrix of $(X, Y)$, before the selection happens.

The proof consists of the following main steps. First, we establish the rank structure when each selection variable $Y_i$ is Gaussian and is conditioned on a single fixed value (i.e., each $\mathcal{Y}_i$ is a singleton subset of $\mathbb{R}$). Then, we extend the result to the case where the selection noise terms $\epsilon_i$ are still Gaussian but $\mathcal{Y}_i$ may contain multiple values. Finally, we generalize to the case where the selection noise terms $\epsilon_i$ are not necessarily Gaussian.

**Step 1: Rank structure under pointwise Gaussian selection.**   Suppose $\epsilon_i$ is Gaussian and each $\mathcal{Y}_i = \{y_i\}$ is a singleton. Let $\Sigma$ denote the joint covariance matrix of $(X, Y)$. Since $(X, Y)$ is jointly Gaussian, the conditional covariance of $X \mid Y = y$ at all different values of $y$ can be expressed as a fixed term:

$$\mathrm{Cov}(X \mid Y = y) = \Sigma_{X,X} - \Sigma_{X,Y}\Sigma_{Y,Y}^{-1}\Sigma_{Y,X}.$$

Now we show how the ranks of submatrices in the conditional covariance matrix $\mathrm{Cov}(X \mid Y = y)$ correspond to those in the original global covariance matrix $\Sigma$. For any subsets $A, B \subset X$, consider the original covariance submatrix:

$$\Sigma_{A \cup Y, B \cup Y} = \begin{bmatrix} E & G \\ F & H \end{bmatrix},$$

where $E = \Sigma_{A,B}$, $G = \Sigma_{A,Y}$, $F = \Sigma_{Y,B}$, $H = \Sigma_{Y,Y}$. We then have

$$\mathrm{rank}\left(\begin{bmatrix} E & G \\ F & H \end{bmatrix}\right) = \mathrm{rank}\left(\begin{bmatrix} I & -GH^{-1} \\ 0 & I \end{bmatrix}\begin{bmatrix} E & G \\ F & H \end{bmatrix}\right)$$
$$= \mathrm{rank}\left(\begin{bmatrix} E - GH^{-1}F & 0 \\ F & H \end{bmatrix}\right)$$
$$= \mathrm{rank}\left(E - GH^{-1}F\right) + |Y|,$$

where the first step is due to multiplying an invertible unit upper triangular matrix, and the second step is because that $H$ is invertible, and thus upper rows (containing 0 at $H$ columns) must be linearly independent with lower rows.

In it, $E - GH^{-1}F$ is $\Sigma_{A,B} - \Sigma_{A,Y}\Sigma_{Y,Y}^{-1}\Sigma_{Y,B}$, which is exactly the covariance $\Sigma_{A,B}^{(\mathcal{S})}$ in the selection-biased data. This completes the proof of this part, showing that $\mathrm{rank}(\Sigma_{A,B}^{(\mathcal{S})}) = \mathrm{rank}(\Sigma_{A \cup Y, B \cup Y}) - |Y|$.

Note that in this part of pointwise selection, the above rank equality actually holds for all joint Gaussian $X, Y$ random variables, not necessarily requiring that $Y$ are outcome-dependent selection response variables–$Y$ can also be $X$'s causes or ancestors. However, for the following multi-valued selection sets, we will explicitly require that $Y$ are $X$'s effects.

**Step 2: Extension to multi-valued Gaussian selection sets.** Now consider the case where each $Y_i$ is Gaussian and is restricted to a measurable subset $\mathcal{Y}_i \subset \mathbb{R}$, and let $\mathcal{Y} = \mathcal{Y}_1 \times \cdots \times \mathcal{Y}_k$. The conditional covariance of $X \mid Y \in \mathcal{Y}$ is given by the law of total covariance:

$$\mathrm{Cov}(X \mid Y \in \mathcal{Y}) = \mathbb{E}[\mathrm{Cov}(X \mid Y) \mid Y \in \mathcal{Y}] + \mathrm{Cov}(\mathbb{E}[X \mid Y] \mid Y \in \mathcal{Y}).$$

We analyze each term:

- Due to Gaussianity, $X \mid Y = y$ is Gaussian with constant covariance under different $y$ values, and thus we have:

$$\mathbb{E}[\mathrm{Cov}(X \mid Y) \mid Y \in \mathcal{Y}] = \Sigma_{X,X} - \Sigma_{X,Y}\Sigma_{Y,Y}^{-1}\Sigma_{Y,X}.$$

- The conditional mean is linear:

$$\mathbb{E}[X \mid Y] = \mu_X + P(Y - \mu_Y), \quad \text{where } P := \Sigma_{X,Y}\Sigma_{Y,Y}^{-1}.$$

Thus,

$$\mathrm{Cov}(\mathbb{E}[X \mid Y] \mid Y \in \mathcal{Y}) = P\,\mathrm{Cov}(Y \mid Y \in \mathcal{Y})P^\top.$$

Putting this together:

$$\mathrm{Cov}(X \mid Y \in \mathcal{Y}) = \Sigma_{X,X} - P\Sigma_{Y,Y}P^\top + P\,\mathrm{Cov}(Y \mid Y \in \mathcal{Y})P^\top.$$

Let $A, B \subset X$. Then the corresponding submatrix can be decomposed to two terms:

$$\Sigma_{A,B}^{(\mathcal{S})} = M_{A,B} + K_{A,B},$$

where let $P_A$ and $P_B$ be the submatrices of $P$ indexed by rows of $A$ and $B$, we have:

$$M_{A,B} := \Sigma_{A,B} - P_A\Sigma_{Y,Y}P_B^\top, \quad K_{A,B} := P_A\,\mathrm{Cov}(Y \mid Y \in \mathcal{Y})P_B^\top.$$

Note that $M_{A,B}$ is the same matrix appearing in the pointwise selection case. So we are now to show that adding the correction term $K_{A,B}$ does not alter the rank of $M_{A,B}$.

Write $Y = \beta X + \epsilon$, where $\beta \in \mathbb{R}^{k \times |X|}$ is the loading matrix of the selection response variables. We then have that in $P$, the term $\Sigma_{X,Y}$ equals $\Sigma_{X,X}\beta^\top$. Thus, the columns of $P \in \mathbb{R}^{|X| \times k}$ lie in the column space of $\Sigma_{X,X}$, i.e., images $\mathrm{Im}(K_{A,B}) \subseteq \mathrm{Im}(\Sigma_{A,Y}) \subseteq \mathrm{Im}(\Sigma_{A,X})$. Since also both $M_{A,B}$ and $K_{A,B}$ lie within the row space of $\Sigma_{A \cup Y, B \cup Y}$, we have:

$$\mathrm{rank}(\Sigma_{A,B}^{(\mathcal{S})}) = \mathrm{rank}(M_{A,B} + K_{A,B}) \leq \mathrm{rank}(M_{A,B}) = \mathrm{rank}(\Sigma_{A \cup Y, B \cup Y}) - |Y|.$$

That is, the correction term $K_{A,B}$, although nonzero, cannot increase the rank. Further, since each $\mathcal{Y}_i$ is a proper subset of $\mathbb{R}$, i.e., not all values are admissible, $\Sigma_{Y,Y}$ and $\mathrm{Cov}(Y \mid Y \in \mathcal{Y})$ will not cancel each other under generic assumption, and thus the equality holds, which completes the proof for this part, showing $\mathrm{rank}(\Sigma_{A,B}^{(\mathcal{S})}) = \mathrm{rank}(\Sigma_{A \cup Y, B \cup Y}) - |Y|$.

**Step 3: Extension to non-Gaussian selection noise.** Our goal is to prove that $\mathrm{rank}\left(\mathrm{Cov}(X_A, X_B \mid Y \in \mathcal{Y})\right) = \mathrm{rank}\left(\Sigma_{A \cup Y, B \cup Y}\right) - k$ holds, even when for each $Y_i = \beta_i^\top X + \epsilon_i$, $\epsilon_i$ may be non-Gaussian, as long as $X$ themselves are still joint Gaussian. We first note that the law of total covariance still holds:

$$\mathrm{Cov}(X \mid Y \in \mathcal{Y}) = \mathbb{E}[\mathrm{Cov}(X \mid Y) \mid Y \in \mathcal{Y}] + \mathrm{Cov}(\mathbb{E}[X \mid Y] \mid Y \in \mathcal{Y}),$$

where the since the linear structure still holds in the global generating process, we still have:

$$\Sigma_{X,Y} = \Sigma_{X,X}\beta^\top, \qquad \mathbb{E}[X \mid Y] = \mu_X + P(Y - \mu_Y), \quad \text{where } P := \Sigma_{X,Y}\Sigma_{Y,Y}^{-1}.$$

The only difference is that, now conditioning on each singleton $y$ value, the covariance term $\mathrm{Cov}(X|Y = y)$ is not a constant anymore. However, since $X$ themselves are still Gaussian, conditioning $X|Y = y$ is equivalent to restrict the noise terms $\epsilon = y - \beta X$, where the right hand side is Gaussian. Therefore, with $X$ independent to $\epsilon$, this imposes affine constraints on $X$, which are still linear in $X$. Since the rank reductions are exactly due to those affine constraints, we have that still

$$\mathrm{rank}(\mathbb{E}[\mathrm{Cov}(A, B \mid Y) \mid Y \in \mathcal{Y}]) = \mathrm{rank}(\Sigma_{A \cup Y, B \cup Y}) - |Y|.$$

Then, using the similar argument as in Step 2, we have the final rank equality, concluding the proof.

Note that when $X$ is non-Gaussian, even when $\epsilon$ is Gaussian, the above properties do not hold, because in that case, $X|Y = y$ is no longer an affine transformation of a Gaussian, and $\mathrm{Cov}(X|Y = y)$ may reflect other nonlinear constraints and have higher rank than expected. $\qquad\square$

## B  Related Work

In this section, we provide a comprehensive review of the relevant literature, focusing on two key areas: latent variable causal discovery and causal discovery or causal inference under selection bias.

**Statistical tools for latent variable causal discovery**   As discussed in §1, standard conditional independence (CI)-based approaches to causal discovery fail to provide sufficient information for recovering latent structures. To address this limitation, a range of alternative statistical tools have been developed, typically by introducing additional parametric or structural constraints. These include rank constraints (Sullivant et al., 2010), which generalize the Tetrad representation theorem from Spirtes et al. (2000) and provide algebraic conditions on covariance matrices; equality constraints derived from Gaussian structural equation models that even has rank constraints as a subclass (Drton, 2018); and high-order moment constraints (Xie et al., 2020; Adams et al., 2021; Robeva & Seby, 2021; Dai et al., 2022; 2024b; Chen et al., 2024), which exploit non-Gaussianity for identifiability. Additionally, matrix decomposition methods (Anandkumar et al., 2013), copula-based constraints (Cui et al., 2018), and mixture oracles (Kivva et al., 2021) were also developed.

**Constraints in linear Gaussian structural equation model**   Among the various statistical tools, rank constraints and their associated graphical criteria are particularly well-known. Recent work has extended these constraints to nested rank constraints, which characterize additional algebraic polynomial varieties in covariance matrices beyond zero-determinant conditions (Drton et al., 2020). These extensions relate to e.g., the Verma constraint, originally formulated in the nonparametric setting (Pearl & Verma, 1995) and later explored with graphical criteria (Evans & Didelez, 2015; Richardson et al., 2023; Bhattacharya & Nabi, 2022). In the context of conditional distributions, the most closely related work is Silva & Shimizu (2017), which primarily aims to validate instrumental variables, rather than establishing a more general graphical criterion for causal structure recovery in the linear non-Gaussian setting.

**Latent variable causal discovery methods**   Building on these statistical tools, a variety of latent variable causal discovery algorithms have been proposed. Many of these methods fall within the constraint-based framework, leveraging CI tests and algebraic constraints to infer causal relations. Notable examples include approaches based on rank or tetrad constraints (Silva et al., 2003; 2006; Silva & Scheines, 2004; Choi et al., 2011; Kummerfeld & Ramsey, 2016; Huang et al., 2022; Dong et al., 2023; 2024). While the majority of these methods fall within the constraint-based paradigm, recent efforts have attempted to formalize score-based methods for latent causal discovery (Jabbari et al., 2017; Ng et al., 2024).

**Causal modeling with selection bias**   Classical approaches to deal with selection bias primarily fall into two categories: (1) nonparametric methods that leverage selection mechanisms to correct for selection effects, and (2) parametric approaches that introduce explicit selection models into causal inference frameworks.

For the purpose of causal discovery, following the foundational work of the FCI algorithm, nonparametric methods have been developed to identify causal relations using conditional independence constraints (Hernán et al., 2004; Tillman & Spirtes, 2011; Evans & Didelez, 2015; Versteeg et al., 2022). There are recent works that deal with selection bias in interventional studies (Dai et al., 2025) and in sequential data (Zheng et al., 2024; Qiu et al., 2024). Several parametric approaches have been developed for bivariate causal orientation (Zhang et al., 2016; Kaltenpoth & Vreeken, 2023).

For the purpose of causal inference (bias adjustment), the nonparametric perspective builds on the graphical representation of selection mechanisms, as introduced in the selection diagram framework (Bareinboim et al., 2014; Bareinboim & Pearl,

2012; Bareinboim & Tian, 2015; Bareinboim & Pearl, 2016; Bareinboim et al., 2022), which characterizes conditions under which causal effects remain identifiable despite selection bias. Subsequent works extended this approach by developing testable implications of selection mechanisms (Correa et al., 2019) and providing adjustment criteria. From a parametric perspective, selection bias has been studied extensively in economics (Heckman, 1977; 1990; Robins et al., 2000).

Structure learning and inference results can follow different methodological directions depending on the problem setting. A particularly relevant area is the study of data missingness, which shares similarities with selection bias. For instance, in cases involving *self-masking* missingness, the true data distribution and model parameters may be unidentifiable, making causal inference infeasible (Mohan et al., 2013; Mohan, 2018). However, the causal structure may still be recoverable (Dai et al., 2024a).

## C  Supplementary Experimental Details and Results

### C.1  Additional Result on Synthetic Data

The difference of Edgemarks with non-Gaussian error terms is shown in Figure. 5. We also report the structural Hamming distance (SHD) of the skeleton to capture the method's ability to recover the structure up to the Markov equivalence classes (represented as PAGs). The result for Gaussian and Exponential error terms are shown in Figure. 7.

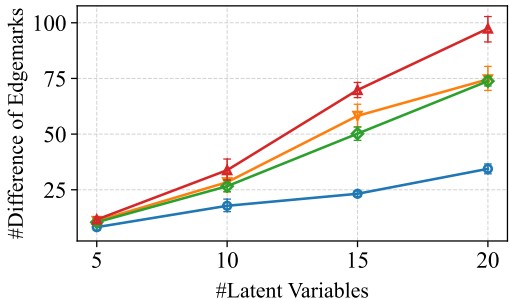

*Figure 5.* Difference of Edgemarks with exponential error terms.

### C.2  Additional Result on Real-world Data

To ensure a diverse and representative evaluation, Australia and India were selected as part of our experiment to encompass all continents and include more developing countries. The results, as illustrated in Figure 8, indicate a potential selection bias related to social norms, which aligns with the findings from *Germany* (Figure 3c). This suggests that the survey design or respondent characteristics may systematically favor certain perspectives on social norms. Additionally, the results reveal a potential selection bias concerning happiness, as shown in Figure 9. This bias can be interpreted as an overrepresentation of individuals who are happier and healthier—both mentally and physically—within the survey sample.

### C.3  World Value Survey Dataset Details

The main research instrument of the World Value Survey project is a representative comparative social survey which is conducted globally every 5 years. Extensive geographical and thematic scope, free availability of survey data and project findings for broad public turned the WVS into one of the most authoritative and widely-used cross-national surveys in the social sciences. At the moment, WVS is the largest non-commercial cross-national empirical time-series investigation of human beliefs and values ever executed. In addition, WVS is the only academic study which covers the whole scope of global variations, from very poor to very rich societies in all world's main cultural zones. The WVS has over the years demonstrated that people's beliefs play a key role in economic development, the emergence and flourishing of democratic institutions, the rise of gender equality, and the extent to which societies have effective government. The full list of 13 variables that we focus on in this study, with their corresponding indicators, are as follows:

- Social Values, Norms, Stereotypes (Q1-Q45)

- Happiness and Wellbeing (Q46-Q56)

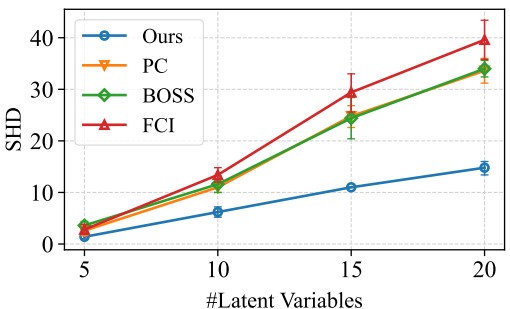

*Figure 6.* SHD results with Gaussian error terms.

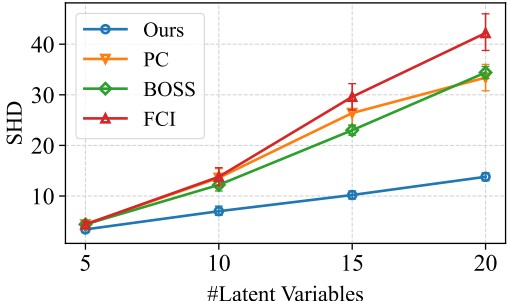

*Figure 7.* SHD results with Exponential error terms.

- Social Capital, Trust and Organizational Membership (Q57-Q105)

- Economic Values (Q106-Q111)

- Perceptions of Corruption (Q112-Q120)

- Perceptions of Migration (Q121-Q130)

- Perceptions of Security (Q131-Q151)

- Index of Postmaterialism (Q152-Q157)

- Perceptions about Science and Technology (Q158-Q163)

- Religious Values (Q164-Q175)

- Ethical Values (Q176-Q198)

- Political Interest and Political Participation (Q199-Q234)

- Political Culture and Political Regimes (Q235-Q259)

We also present below some examples of the raw questions in the section of *Happiness and Wellbeing*:

- Q46: *Feeling of happiness: Taking all things together, would you say you are?*

- Q47: *State of health (subjective): All in all, how would you describe your state of health these days?*

- Q48: *How much freedom of choice and control: Please use a scale where 1 means "none at all" and 10 means "a great deal" to indicate how much freedom of choice and control you feel you have over the way your life turns out.*

- Q49: *Satisfaction with your life: All things considered, how satisfied are you with your life as a whole these days?*

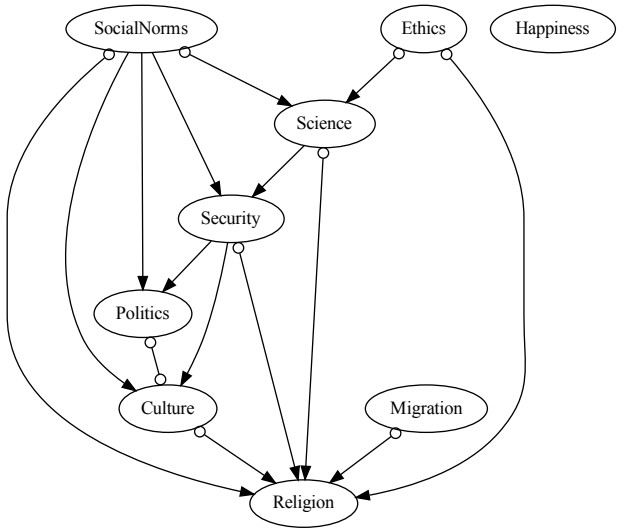

*Figure 8.* Output PAG in *India*.

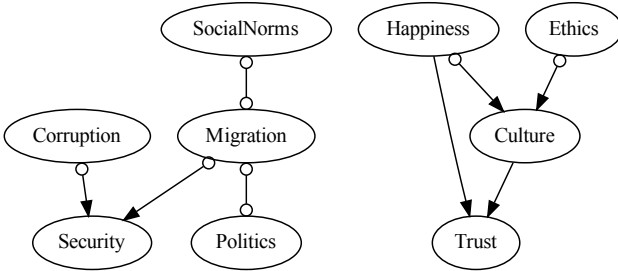

*Figure 9.* Output PAG in *Australia*.

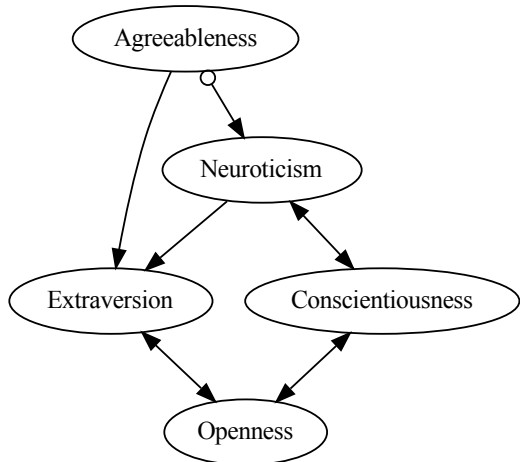

*Figure 10.* Output PAG for *BIG5*.

- Q50: *Satisfaction with financial situation of household: How satisfied are you with the financial situation of your household? If '1' means completely dissatisfied and '10' means completely satisfied, where would you place your satisfaction?*

- Q51: *Frequency you/family (last 12 months): Gone without enough food to eat.*

- Q52: *Frequency you/family (last 12 months): Felt unsafe from crime in your own home.*

- Q53: *Frequency you/family (last 12 months): Gone without needed medicine or treatment that you needed.*

- Q54: *Frequency you/family (last 12 months): Gone without a cash income.*

- Q55: *Frequency you/family (last 12 months): Gone without a safe shelter over your head.*

- Q56: *Standard of living comparing with your parents: Comparing your standard of living with your parents' standard of living when they were about your age, would you say that you are better off, worse off, or about the same?*

## C.4    Data-preprocessing

In the WVS dataset, the data preprocessing consists of two main components: sample choice and measurement variable choice. For sample choice, we remove those with data missingness in entries (e.g., some questions left unanswered or "preferred not to answer"). For measurement variable choice, though the questions are already categorized into different aspects, to ensure that the rank method works, we find subsets from each latent variable's corresponding measurements that share a rank-1 structure against others, i.e., they indeed appear as masurements to a same latent variable.

