# OpenReview forum: "Latent Variable Causal Discovery under Selection Bias"
_ICML.cc/2025/Conference — ICML 2025 poster_

### Official Review · Reviewer_KB6X · 2025-02-25

**Overall Recommendation:** 4

**Summary:**

This paper investigates the problem of Latent Variable Causal Discovery in the presence of Selection Bias and proposes a new statistical tool, Generalized Rank Constraints, to simultaneously handle latent variables and selection bias within linear Gaussian models.

## update after rebuttal
I have decided to raise my score from 3 to 4. The authors have provided thorough and thoughtful responses to my concerns. For Q1, they addressed the challenge of distinguishing selection bias from latent confounding by discussing both the limitations of second-order information and the potential of higher-order information with additional assumptions. For Q2, they clarified the computational complexity of the generalized rank constraints compared to conditional independence tests, highlighting the efficiency of their method. For Q3, they acknowledged the omission of the RFCI reference and included it in the revised manuscript. These detailed and constructive responses demonstrate the authors' commitment to addressing feedback and improving the quality of their work, leading me to adjust my score upward.

**Claims And Evidence:**

The claims submitted are supported by clear and convincing evidence.

**Essential References Not Discussed:**

There are some latent variable causal discovery algorithms not discussed and compared in this paper, for example:
[1] Colombo D, Maathuis M H, Kalisch M, et al. Learning high-dimensional directed acyclic graphs with latent and selection variables[J]. The Annals of Statistics, 2012: 294-321.

**Experimental Designs Or Analyses:**

I believe the experimental design or analysis is reasonable and valid.

**Methods And Evaluation Criteria:**

The proposed method and/or evaluation criteria (e.g., benchmark datasets) are meaningful for the problem or application at hand.

**Other Comments Or Suggestions:**

No

**Other Strengths And Weaknesses:**

Strength:
This solves the problem of eliminating the influence of selection bias to discover hidden variables and restore causal relationships when hidden variables and selection bias exist at the same time.
Weakness:
1. The paper mentions that certain latent variable structures and selection bias structures may be equivalent under rank constraints. How can they be distinguished?
2. There is a lack of discussion on the time complexity comparison between the generalized rank constraints method and conditional independence tests.

**Questions For Authors:**

1. The paper mentions that certain latent variable structures and selection bias structures may be equivalent under rank constraints. How can they be distinguished?
2. There is a lack of discussion on the time complexity comparison between the generalized rank constraints method and conditional independence tests.

**Relation To Broader Scientific Literature:**

This solves the problem of eliminating the influence of selection bias to discover hidden variables and restore causal relationships when hidden variables and selection bias exist at the same time.

**Theoretical Claims:**

No

---

> ### Author Rebuttal · Authors · 2025-04-01
>
> We are grateful for the reviewer's insightful comments and suggestions. Please see below for our response.
>
> ---
>
>
> **(Q1)** The reviewer wonders how one might distinguish between selection bias and latent confounding in practice.
>
> **A:** Thank you for this insightful question. We try to address it from two complementary perspectives:
>
> 1. **Using only second-order information (covariances), this is a hard problem:**
>    - Distinguishing selection bias from latent confounding based **solely on rank constraints is, in general, a hard problem.** Ideally, one would need to characterize the rank equivalence class–the set of all graphs (with potentially different latent and selection variables) that entail the same rank constraints. Then, the presence of selection or confounding could only be confirmed when all rank-equivalent graphs contain them.
>    - However, a general characterization of rank equivalence **remains open and challenging,** even in the setting without selection bias. Due to space limit, please kindly refer to our response to Reviewer nj2K's Q1 for detailed discussion.
>    - Moreover, even with a full characterization of rank equivalence, distinguishing between selection and confounding–as in our Section 3.3–may still **require algorithmic reasoning** over sets of rank constraints rather than straightforward graphical patterns. This is similar to the setting with only CI constraints. For instance, in our Example 4, the presences of latent variables or selection are implied by FCI's output, not by any obvious graphical patterns.
>
>
> 2. **With higher-order information and additional parametric assumptions, certain distinctions may become feasible:**
>    - Let us consider the example of two measured variables. Whether they are confounded ($X_1 \leftarrow L \rightarrow X_2$), directly causal related ($X_1 \rightarrow X_2$), or selected ($X_1 \rightarrow Y \leftarrow X_2$) are indistinguishable by rank or CI constraints, as all three have $X_1 \not \perp X_2$.
>    - However, they may become distinguishable with more parametric assumptions. For the direct causal effect, for instance, in a linear non-Gaussian setting, it can be distinguished from the other two cases via independence of regression residuals.
>    - For the case of selection bias, suppose the selection acts through truncation, and the scatterplot of observed variables may exhibit a clear truncation pattern (e.g., as in Figure 2 in our draft). Reproducing this pattern under the other two cases **would require more complex**–and arguably less plausible–functional forms, often by reverse-engineering rather than natural functions.
>    - Hence, with a proper **definition of simplicity and the preference for it**, the model can be further identified.
>
> Note that observations in (2) fall outside the scope of this work, which serves mainly as a (first) proof of concept to show that rank constraints alone can remain informative under selection bias.
>
> However, we sincerely appreciate the reviewer's question, which points to an important future direction: exploring which additional assumptions are needed, and **how the above two lines can be combined** to enable a more practical identification of selection bias and latent confounding in real-world scenarios.
>
> ---
> **(Q2)** The reviewer asks for the time complexity comparison between the generalized rank constraints and conditional independence tests.
>
> **A:**  Thank you for the question. Please let us note that the generalized rank constraints do not change the computational complexity of the statistical tests themselves–they simply **provide a characterization** of the covariance terms for recovering the graph.
>
> Specifically, testing rank constraints (e.g., via canonical correlation analysis) and testing for conditional independence (e.g., via Fisher's Z-test) both involve computations on the covariance matrix. The underlying operations–matrix inversion, or eigenvalue decomposition–are **of comparable complexity.**
>
> As for the algorithmic procedure used to recover the graphs, ours follows the same structure as FCI. Hence, like FCI, our algorithm has a worst-case complexity exponential in the number of observed variables. However, if the underlying graph is sparse–a common and reasonable assumption–**the runtime becomes polynomial.**
>
> We also report our empirical running times. Though with a same complexity, our method is **consistently the fastest** among competitors, possibly thanks to some implementation speedup. Due to space limit, please kindly refer to our response to Reviewer nKfC's Q1 for the results.
>
> ---
>
> **(Q3)** The reviewer notes that a relevant work (RFCI) was not discussed.
>
> **A:** We thank the reviewer for pointing out this relevant reference [[C+12]](https://tinyurl.com/4mw7h5ku). We have now included it in the updated manuscript and revised our discussion accordingly.
>
>
> ---
>
> Once again, we thank the reviewer for the insightful feedback, and hope the questions have been properly addressed.

---

> > ### Comment · Reviewer_KB6X · 2025-04-02
> >
> > Thank you for addressing the feedback. After re-assessing the manuscript and evaluating the revisions, I have decided to elevate the score from 3 to 4.

---

> > > ### Author Response · Authors · 2025-04-08
> > >
> > > We are delighted to know that your questions have been well addressed. We sincerely appreciate your thoughtful feedback and recognition of our work. Thank you.

---

### Official Review · Reviewer_nKfC · 2025-03-10

**Overall Recommendation:** 3

**Summary:**

The authors address the problem of causal discovery with latent confounders when the data has a selection bias. Here, the authors address this via generalized rank constraints that extend beyond conditional independencies. The ranks of covariance submatrices in biased data can reveal information about the latent causal structure and the selection mechanism. The proposed method has been evaluated in artificial and real-world data.

**Claims And Evidence:**

Under the given assumptions (linear and Gaussian), Theorem 1 supports the claim that ranks preserve structural information under selection bias. However, distinguishing between latent confounding and selection bias is only discussed through examples.

**Essential References Not Discussed:**

N/A

**Experimental Designs Or Analyses:**

The experiments are fair, but use a simplistic selection mechanism and the real-world data lacks quantitative validation.

**Methods And Evaluation Criteria:**

The run experiments make sense, although real-world validations rely heavily on qualitative assessments.

**Other Comments Or Suggestions:**

See questions.

**Other Strengths And Weaknesses:**

- Good motivation
- Fair discussion of related work and pointing out the novel aspect
- Clear notation and figures

- Limited to linear Gaussian models and linear selection mechanism
- Distinguishing between latent confounding and selection bias is unclear

**Questions For Authors:**

The paper addresses an interesting problem and is novel in this regard. While the theoretical contribution is great (even though it requires some limiting assumptions), the experiments could be more insightful. Some questions:

- It is unclear how robust your approach is against a violation of linearity and the Gaussianity assumption. Experiments with data that explicitly violates your main assumptions would be insightful.
- In practice, how could one determine if selection bias is present in a dataset versus just latent confounding?
- The complexity of the proposed method is unclear. It generally appears to be fast (especially compared to PC/FCI etc.). Can the authors briefly comment on this?
- How robust is the rank determination under small sample sizes, do you have some insights on this?

**Relation To Broader Scientific Literature:**

Fair discussion of how related literature is lacking in the area of latent confounders under selection bias. While some discovery methods that support latent variables are discussed, it is only a small (but sufficient) selection.

**Theoretical Claims:**

Theorem 1 appears to be sound and Proposition 3 directly follows from applying the theorem to one-factor models.

---

> ### Author Rebuttal · Authors · 2025-04-01
>
> We sincerely thank the reviewer for the insightful comments. Below, we provide detailed response.
>
> ---
> **(Q1)** The reviewer suggests several additional experiments.
>
> **A:**  Thank you for the constructive suggestions. In light of them, we conducted new experiments to evaluate our method under violations of **Gaussianity** and **linearity**, under **small sample sizes**, and regarding **time complexity**. Results are summarized below and detailed in [this anonymous link](https://tinyurl.com/ydmznrxm).
>
> 1. **Non-Gaussianity**: We simulate linear SEMs with noise types: Gaussian, Exponential, Laplace, Uniform, and Gumbel. These choices test the sensitivity to skewness, tail behavior, and asymmetry. The numbers of edgemark differences on 10 and 20 nodes and 5 random seeds (same below) are reported. Despite the violations, our method consistently outperforms others:
>    |#Nodes|10||||20||||
> |-|-|-|-|-|-|-|-|-|
> |**Noise**|Ours|PC|BOSS|FCI|Ours|PC|BOSS|FCI|
> |Gaussian|**19.0±8.3**|28.2±9.1|27.6±12.6|35.6±12.2|**39.4±6.4**|76.0±7.2|65.4±4.3|88.8±6.9|
> |Exponential|**19.8±7.8**|27.4±8.1|28.4±10.4|35.0±11.8|**40.6±5.8**|69.2±6.2|60.0±9.6|80.2±11.4|
> |Laplace|**18.0±7.9**|28.4±7.9|22.2±6.4|33.2±11.3|**39.6±3.0**|70.0±11.2|62.8±11.0|84.8±8.5|
> |Uniform|**17.0±8.6**|28.8±9.6|30.6±11.3|34.6±11.9|**35.4±6.9**|64.0±9.8|72.6±12.7|77.4±10.7|
> |Gumbel|**18.4±8.1**|25.8±6.2|24.6±7.4|33.8±10.5|**38.8±6.0**|69.6±10.3|64.0±10.8|80.4±11.7|
>
> 2. **Nonlinearity**: We simulate additive noise SEMs with functions: Linear, Leaky ReLU, Tanh, Cubic, Quadratic, and Since. All but the last two are monotonic. We do see a performance drop on the last two, while comparing to other methods, ours remains the best:
>     |#Nodes|10||||20||||
> |-|-|-|-|-|-|-|-|-|
> |**Function**|Ours|PC|BOSS|FCI|Ours|PC|BOSS|FCI|
> |Linear|**19.0±8.3**|28.2±9.1|27.6±12.6|35.6±12.2|**39.4±6.4**|76.0±7.2|65.4±4.3|87.6±4.2|
> |Leaky ReLU|**20.0±7.7**|28.4±8.6|27.6±12.5|32.6±11.5|**42.4±6.1**|69.6±7.2|71.6±10.0|87.0±7.1|
> |Tanh|**20.8±6.8**|28.8±9.0|24.0±8.8|30.0±13.3|**38.2±6.4**|66.6±9.1|50.6±9.8|77.8±4.6|
> |Cubic|**25.0±4.3**|28.0±5.0|27.8±4.7|34.0±3.6|**54.0±5.7**|75.8±12.4|64.0±8.8|81.5±13.4|
> |Quadratic|**25.8±3.5**|26.2±3.7|28.6±4.8|27.8±5.0|**55.2±3.6**|61.4±8.8|59.6±10.4|70.6±5.7|
> |Sine|27.4±2.7|28.2±4.4|**25.8±6.4**|30.8±6.2|**47.6±6.5**|60.0±9.0|48.8±6.6|66.0±9.6|
>
> 3. **Small sample sizes**: Using linear Gaussian SEMs, we vary sample size. With fewer samples, our method continues to detect low-rank patterns reliably and outperforms others:
>    |#Nodes|10||||20||||
>  |-|-|-|-|-|-|-|-|-|
>  |**#Samples**|Ours|PC|BOSS|FCI|Ours|PC|BOSS|FCI|
>  |100|**24.6±4.5**|24.8±7.7|25.0±5.7|26.0±5.4|**53.8±4.2**|54.8±5.6|55.6±6.4|56.4±3.7|
>  |500|24.0±6.9|**23.8±6.5**|27.4±8.3|29.6±7.5|**46.2±11.7**|55.4±10.1|48.8±10.3|60.2±11.0|
>  |1000|**22.0±6.2**|26.8±8.1|22.4±7.5|27.4±8.2|**53.0±5.9**|62.2±7.0|61.4±6.2|74.2±10.1|
>
> 4. **Time complexity**: Our algorithm follows the same structure as FCI. Hence, like FCI, it has a worst-case complexity exponential in the number of variables. However, if the graph is sparse–a common and reasonable assumption–the runtime becomes polynomial. We report running time in ms below:
>    |#Nodes|10||||20||||
>  |-|-|-|-|-|-|-|-|-|
>  |**#Samples**|Ours|PC|BOSS|FCI|Ours|PC|BOSS|FCI|
>  |100|**12±1**|479±15|453±19|447±15|**43±4**|488±29|462±12|493±41|
>  |1000|**28±8**|493±19|475±35|500±30|**110±31**|542±29|503±13|606±53|
>  |10000|**110±48**|629±31|604±42|665±54|**434±113**|976±106|776±32|4364±3554|
>
> ---
> **(Q2)** The reviewer wonders how one might distinguish between selection bias and latent confounding in practice.
>
> **A:** Thank you for this insightful question. Below, we try to briefly address it from two complementary views. Due to space limit, for a more detailed discussion, please kindly refer to our response to Reviewer KB6X's Q1.
>
> 1. **Using only second-order information (rank in covariances), this is a hard problem**:
>    - Ideally, **one would need to characterize the rank equivalence class**, which is an open and challenging problem (see, e.g., our response to Reviewer nj2K's Q1).
>    - Even with a full characterization, distinguishing the two may still **require algorithmic reasoning** over constraint sets, rather than any obvious graphical patterns.
>
> 2. **With higher-order information and parametric assumptions, some distinctions become feasible**:
>    - Even for two rank-equivalent graphs, data generated from one graph with a natural SEM may require a much more complex, unnatural SEM to reproduce under the other graph.
>    - Such distinctions with a **preference for simplicity can guide model selection.**
>
> Though (2) is beyond this work's scope, we sincerely appreciate the reviewer's question that highlights a valuable direction: seeing how the two lines can be merged to better identify selection and confounding in practice.
>
> ---
> We want to thank the reviewer again for all the valuable feedback, and we hope the reviewer's questions are properly addressed.

---

### Official Review · Reviewer_6VTP · 2025-03-10

**Overall Recommendation:** 4

**Summary:**

In this paper, the authors propose the use of rank constraints to infer causal structure of the latent variables underlying a set of measurements, when these latent variables are themselves subject to selection bias (e.g. more conscientious individuals are more likely to fill out a full Big 5 questionnaire).

### update after rebuttal ###

I thank the authors for their answers. My assessment remains positive.

**Claims And Evidence:**

Yes.

**Essential References Not Discussed:**

The most relevant papers are discussed.

**Experimental Designs Or Analyses:**

The setup of the empirical evaluation is sound, including experiments on synthetic and real-world data. As the authors primarily compare to basic rather than state-of-the-art methods, there is a risk for an illusion of progress.

**Methods And Evaluation Criteria:**

Yes.

**Other Comments Or Suggestions:**

None.

**Other Strengths And Weaknesses:**

- Strengths
The approach is novel and interesting.
Few methods can effectively deal with both latent variables and selection at the same time.
The paper is clear and straightforward.

- Weaknesses
It's unclear how the results here can be used to deal with cases where we don't know how many latent variables there are
It's not quite clear to me how Theorem 1 can be used when we don't know what the selection variables are

**Questions For Authors:**

- Can you explain why in Fig. 3, the graph for China contains more nodes than for Canada and Germany?
- In the experiment of the Big 5, can you explain to what extent the results depend on the "objective" existence of the measured traits? Big 5 often involve various preprocessing steps (such as varimax rotation), how do we know our results are not due to these preprocessing steps?
- In Definition 4 we assume that the parents of each $Y_i$ are among the latent variables $L$. Does this matter, when the paths from $L \to X \to Y$ would also all be linear?
- In Fig. 4 (or in a followup in the Appendix), it would be interesting to see the kinds of errors made by the algorithm. Does it mostly miss selection, confounding, or something else?

**Relation To Broader Scientific Literature:**

The paper related its key contributions well with regard to the literature.

**Theoretical Claims:**

I checked the proofs and they seem correct.

---

> ### Author Rebuttal · Authors · 2025-04-01
>
> We appreciate the reviewer's constructive comments. Please see below for our response.
>
> ---
> **(Q1)** The reviewer wonders how to identify latent structure and selection bias when the structural assumption of one-factor models are violated–such as when the number of latent variables is unknown, or when selection acts on observed variables as $L \to X \to Y$.
>
> **A:** Thank you for this insightful question. Moving beyond structural assumptions has long been a goal–but also a challenge–for rank-based methods. Below, we address this from two complementary perspectives: why the problem is inherently hard with rank constraints alone, and how additional information beyond ranks may help.
>
> 1. **Using only second-order information, this is a hard problem:**
>    - Distinguishing selection from confounding via ranks in arbitrary graphs requires characterizing the **rank equivalence class**–the set of all graphs (with possibly different latent and selection variables) that entail the same rank constraints.
>    - However, this characterization **remains open and challenging, even without selection.** Due to page limit, please kindly see our response to Reviewer nj2K's Q1 for details.
>    - In short, despite four decades of work on rank-based methods, no such characterization has been developed, which is why current methods, including ours, typically rely on structural assumptions (e.g., one-factor models).
>    - Even with such a characterization, distinguishing selection and confounding may still **require algorithmic reasoning** over constraint sets rather than direct graphical patterns (see Example 4).
>
> 2. **With higher-order information and additional parametric assumptions, certain distinctions may become feasible:**
>    - Consider two measured variables. Whether they are confounded ($X_1 \leftarrow L \to X_2$), directly causal ($X_1 \to X_2$), or selected ($X_1 \to Y \leftarrow X_2$) is indistinguishable by rank or CI constraints, since all imply $X_1 \not \perp X_2$.
>    - But they may be distinguishable under additional assumptions. For instance, in linear non-Gaussian models, direct causation can be identified via residual independence.
>    - For selection bias, suppose it acts through truncation, scatterplots may show clear cut-offs (see Fig. 2). Reproducing such patterns under the other scenarios would require **more complex**–and arguably less plausible–functional forms. With a **preference for simplicity**, such patterns can guide model selection.
>
> Though (2) lies beyond this work's scope, we appreciate the reviewer’s question, which points to a valuable direction: merging the two lines to better identify selection and confounding in practice.
>
> ---
> **(Q2)** The reviewer asks for clarification on several experimental details:
>
> - **(Q2.1)** Why in Fig. 3, the graph for China contains more nodes than for Canada and Germany?
>
>   **A:** Thank you for your careful reading. As noted in Sec. 5.2, "some variables are missing in certain countries due to the low response rate."
>
> - **(Q2.2)** In Big 5, to what extent do results depend on the "objective" existence of the measured traits?
>
>   **A:** Although traits are labeled for the personality they are designed for (e.g., "Openness"), **we do not use such side information.** Instead, we cluster traits using rank constraints: under one-factor models, two observed traits share a latent parent iff the rank between them and all others is one. These inferred clusters indeed align well with the labels.
>
> - **(Q2.3)** How do we know our results on Big 5 are not due to preprocessing (e.g., varimax rotation)?
>   **A:** Thank you for raising this. While preprocessing like varimax rotation is common in factor analysis, here we directly use the raw, discrete data from [link](https://tinyurl.com/4brackf6)–**no preprocessing is applied.**
>
>   That said, directly verifying model assumptions is still difficult due to selection-induced distortions. However, from another view, the many low-rank patterns–which would typically be destroyed under model violations–are detected, and the recovered graph appears reasonable. This offers a **partial empirical validation** for model adequacy. See Reviewer nj2K’s Q3 for more.
>
> - **(Q2.4)** The comparisons are mainly to basic methods.
>
>   **A:** We do have included BOSS [[A+23]](https://tinyurl.com/3saupn8x), a recent SOTA outperforming classical constraint and score-based methods. No other latent-variable methods are used, because they require different structural assumptions and the outputs are incomparable.
>
> - **(Q2.5)** It would be interesting to see the kinds of errors made by the algorithm.
>
>   **A:** Thank you–this is a valuable point. In the 20-node setting of Fig. 4, our method greatly improves selection-edge ('--') recovery (F1: 0.12 to 0.77). Though '--' edges are few, their correct detection aids propagation, improving direct-edge ('->') recovery as well (F1: 0.52 to 0.80).
>
> ---
> We want to thank the reviewer again for all the valuable feedback.

---

### Official Review · Reviewer_nj2K · 2025-03-14

**Overall Recommendation:** 4

**Summary:**

This paper extends the rank constraint and t-separation criteria in latent variable causal discovery to scenarios involving selection bias. The authors demonstrate that the rank constraint retains its informativeness even under selection bias, despite the potential invalidation of the linear Gaussian assumption. Additionally, they generalize the t-separation criteria to the augmented selection graph, enhancing its applicability. Leveraging these developed tools, the authors investigate the problem of causal discovery in a one-factor model under conditions of selection bias.

**Claims And Evidence:**

Yes. The two primary theoretical contributions of this paper are Theorem 1 and Proposition 4. Specifically, Theorem 1 extends the t-separation criteria introduced by Sullivant et al. (2010) to scenarios involving selection bias, thereby broadening its applicability. Proposition 4 builds on this extension to address the causal discovery problem in a one-factor model under selection bias. Both results are supported by rigorous, well-constructed proofs and are complemented by intuitive illustrations that enhance their clarity and accessibility.

**Essential References Not Discussed:**

No

**Experimental Designs Or Analyses:**

Yes, the experiment design and analyses are sound.

**Methods And Evaluation Criteria:**

Yes. The major proposal of this paper is that the rank constraint remains valid for causal models under selection bias. The authors have successfully demonstrated their points using intuitive examples and rigorious theorems. It is convincing that the proposed method can work under latent causal discovery with selection bias.

**Other Comments Or Suggestions:**

No

**Other Strengths And Weaknesses:**

A weakness of this paper is the lack of a comprehensive characterization of rank equivalence graph, as also acknowledged by the authors. This can affect the application of their theory to real world causal discovery problems.

**Questions For Authors:**

1. I am particularly interested in Sec. 3.3. While a detailed characterization of the rank equivalence graph is not yet available, can the authors provide some intuition or potential ideas to this problem? Further, is it possible to delevep an algorithm like the FCI to recover the rank equivalent graph? How is this rank equivalent graph related to the causal graph over latent variables (which is our target)?

2. Can the rank-constraint theory be extended beyond linear Gaussian data? (I presume not) How this method work empirically when the linear Gaussian assumption does not hold, e.g., with linear nonGaussian data or nonlinear data?

3. Does the personality trait example in the introduction section satisfies the linear Gaussian model?

**Relation To Broader Scientific Literature:**

This paper extends the rank constrain and t-separation criteria introduced by Sullivant et al. 2010 to cases with selection bias.

**Theoretical Claims:**

Yes, I have checked the correctness of the proofs.

---

> ### Author Rebuttal · Authors · 2025-04-01
>
> We sincerely appreciate the reviewer's constructive comments and helpful feedback. Please see below for our response.
>
> ---
> **(Q1)** The reviewer asks about intuitions for the rank equivalence class.
>
> **A:** Thank you for this constructive question. We fully agree that the following three problems are fundamental:
>
> 1. A graphical criterion to decide rank equivalence,
> 2. A formal object to describe the rank equivalence class,
> 3. An algorithm to enumerate all members of the class.
>
> However, as the reviewer noted, **all three remain open and challenging.** To illustrate the hardness, we may consider e.g., the spider graph and the (2,6)-factor analysis graph, and see how two seemingly so different graphs can still, counterintuitively, be rank equivalent [[SSK10]](https://tinyurl.com/5z8x7td5).
>
> Indeed, despite almost four decades of work on rank-based methods since [[G+86]](https://tinyurl.com/2s48ba2v), no general characterization of rank equivalence has yet been developed, even in settings without selection bias. This is why these methods, including ours, typically rely on structural assumptions (e.g., one-factor models), rather than arbitrary graphs.
>
> That said, we see two potential directions:
>
> 1. **Algebraic direction**: Rank constraints can be seen as encoding flow capacities between node sets. Reformulated, the problem becomes: _Given unknown graph structure, but known flow (e.g., min-cut) values between certain node groups, what can we recover?_ Algebraic geometry may offer tools to approach this.
>
> 2. **Algorithmic direction**: Even without a full characterization, one might still build a sound but incomplete algorithm first, by relaxing existing methods gradually. This also mirrors the trajectory of CI-based methods: while FCI was first introduced in 1993 [[SGS93]](https://tinyurl.com/yyzy32f3), the direct graphical criterion for equivalence came in 2002 [[RP02]](https://tinyurl.com/3afjpm98), the full class characterization in 2008 [[Zha08]](https://tinyurl.com/2s47c6yc), and enumeration algorithms are still under study [[WDZ24]](https://tinyurl.com/yeys53vb).
>
> Finally, even with a full characterization, distinguishing latent variables from selection bias–as in our Section 3.3–may still require **algorithmic reasoning over sets of rank constraints rather than graphical patterns.** This mirrors the CI setting–such as in our Example 4, the presence of latent variables or selection is implied by FCI's output, not by any obvious graphical patterns.
>
> We thank the reviewer again for highlighting this important and exciting problem.
>
> ---
> **(Q2)** The reviewer wonders whether the rank-constraint theory can be extended beyond the linear Gaussian setting.
>
> **A:** This is a very important question. We address it from the following three folds:
>
> 1. **Theoretical scope**: Our results indeed rely on properties of the linear Gaussian model. Specifically, Theorem 1 relies on the closure of Gaussians under point conditioning, so that linear structures in covariances are preserved under selection. This does not hold in general non-Gaussian or nonlinear settings.
>
> 2. **Empirical robustness**: Nonetheless, simulations on both linear non-Gaussian and nonlinear data show that our method remains empirically effective (consistently the best among competitors). Due to space limits, please kindly refer to our response to Reviewer nKfC for results.
>
> 3. **Broader outlook**: Finally, we note that this work serves as a (first) proof of concept showing that ranks can remain informative under selection bias. Just as original linear-Gaussian rank results were extended to, e.g., nonlinear, cyclic [[Spi13]](https://tinyurl.com/25y2wx4s), and discrete models [[Gu25]](https://tinyurl.com/n6jm5jry), we believe the insights here may generalize as well, though the tools may differ.
>
> ---
> **(Q3)** The reviewer wonders whether the personality trait data satisfies the linear Gaussian model.
>
> **A:** Thank you for this insightful question. **Strictly speaking, no**–they are discrete questionnaire responses. However, some literature interprets such ordinal variables as discretized versions of Gaussians, in which case covariance ranks can still be informative [[LN08]](https://tinyurl.com/2423ea4d).
>
> More broadly, **model testing in our setting is difficult:** Even if the data were first generated by a linear Gaussian model, selection can then induce strong non-Gaussianity and nonlinearity. Hence, even with Gaussian tests like Mardia's, we cannot easily tell whether a non-Gaussianity arises from model violation or simply selection distortions.
>
> However, from another view, note that our method detects many low-rank constraints in the Big Five dataset. Since major model violations would typically destroy such low ranks in the covariances, these low ranks, together with the meaningful causal structure recovered from them, offer a **partial empirical validation for model adequacy.**
>
> ---
> Once again, we thank the reviewer for the insightful feedback.

---

### Decision · Program_Chairs · 2025-05-01

**Decision:**

Accept (poster)

**Comment:**

All reviewers agree that this is a solid nethodological contribution to the field of causal discovery; in the presence of latent variables.